# Energy Metabolism in Relation to Diet and Physical Activity: A South Asian Perspective

**DOI:** 10.3390/nu13113776

**Published:** 2021-10-25

**Authors:** Siti N. Wulan, Qaisar Raza, Hera S. Prasmita, Erryana Martati, Jaya M. Maligan, Uma Mageshwari, Itrat Fatima, Guy Plasqui

**Affiliations:** 1Study Program of Food Science and Technology, Department of Agricultural Product Technology, Faculty of Agricultural Technology, Brawijaya University, Malang 65145, Indonesia; herasisca87@ub.ac.id (H.S.P.); erryana_m@ub.ac.id (E.M.); maharajay@ub.ac.id (J.M.M.); 2Department of Food Science and Human Nutrition, Faculty of Biosciences, University of Veterinary and Animal Sciences, Lahore 54000, Pakistan; 3Department of Health Sciences, Faculty of Earth and Life Sciences, VU University Amsterdam, 1081 HV Amsterdam, The Netherlands; 4Department of Food Service Management and Dietetics, Faculty of Home Science, Avinashilingham University, Coimbatore 641043, India; uma_fsmd@avinuty.ac.in; 5Department of Food Science and Technology, Faculty of Life Sciences, University of Central Punjab, Lahore 54000, Pakistan; itrat.fatima@ucp.edu.pk; 6Department of Nutrition and Movement Sciences, School of Nutrition and Translational Research in Metabolism, Maastricht University Medical Center (MUMC+), 6229 HX Maastricht, The Netherlands; g.plasqui@maastrichtuniversity.nl

**Keywords:** energy metabolism, South Asian, diet, physical activity

## Abstract

The prevalence of overweight and obesity is on the rise around the world, not only in the West, but also in Asian countries. South Asian countries in particular are experiencing a rapid increase in overweight and obesity, that coexists with the rapid increase in non-communicable diseases linked to obesity such as diabetes and cardiovascular disease than any other country in Asia. The phenomena observed in South Asian countries are due to the size of the population, the ageing of the population, the high degree of urbanization and the lifestyle changes in favor of increased energy consumption and reduced physical activity. The imbalance between energy consumption and energy expenditure results in the development of a positive energy balance that, over time, accumulates in higher body fat. South Asians were reported to have a more unfavorable body composition with a higher percentage of body fat than Caucasians with an equivalent BMI. Body composition is a major determinant of resting energy expenditure. It has been reported that South Asians have a lower resting energy expenditure than Caucasians with the same BMI. Resting energy expenditure accounts for the majority of total daily energy expenditure and, therefore, plays a crucial role in achieving the balance between energy intake and expenditure.

## 1. Introduction

Overweight and obesity are on the rise globally, not only in developed countries, but also in developing countries that are progressing towards a more industrial way of living such as in Asia. In 1995, the WHO found a greater problem with overweight than underweight in developing countries [1]. The global incidence of overweight and obesity is estimated at 1.3 billion and 573 million people, respectively, by 2030, and 43% and 21% of that number live in Asia [2]. In addition, it has been estimated that the world prevalence of diabetes in adults (20–79 years old) will increase from 6.4% in 2010 to 7.7% in 2030 affecting more than 400 million people [3]. Considering the current trends, the increase is expected to be around 69% in developing countries and 20% in developed countries [3]. By 2030, around 80% of people with diabetes will live in developing countries, of which India and China have the largest proportions [4]. Dissimilar to Western countries, diabetes in Asia has developed more rapidly (3–5-fold increase in 30 years), in people with a younger age (20–64 years old), and a lower BMI [5]. The much larger increase in overweight and obesity as well as non-communicable diseases associated with obesity in developing regions are driven by the growing population, the ageing population, the high degree of urbanization, and the lifestyle changes toward increased energy consumption and reduced physical activity [2,6]. This review addresses the possible determinants of the increased prevalence of obesity and metabolic syndrome in the South Asian population. The obesity in South Asians is characterized by an unfavorable body composition leading to a lower (resting) energy expenditure. The inability of this population to adapt to an increased energy (fat) intake, may be attributed to a lower resting energy expenditure, which makes this population more susceptible to a positive energy balance. Reduced physical activity energy expenditure may exacerbate the accumulation of a positive energy balance and the rapid increase in obesity and metabolic syndrome in the South Asian population. 

## 2. Obesity and Metabolic Syndrome: A South Asian Perspective

The prevalence of obesity and obesity-related diseases in people of South Asian descent (Indian sub-continent) is the highest among Asians [5,7]. It has been reported that adults with type 2 diabetes in India increase by 1.8 million annually [3]. The number continues to rise in both native and migrant South Asians [8,9]. In their home countries, the prevalence is increasing in urban, semi-urban, and rural areas [10]. Metabolic syndrome affects approximately one-third of the urban population in large cities in India [11]. South Asians are not only susceptible to diabetes, they also have an earlier onset, and more adverse and more frequent cardiovascular disease (CVD) than other ethnicities [12]. Several studies have revealed the higher risk of CVD in South Asians is associated with an unfavorable lipid profile or dyslipidemia [9] such as an elevated fasting plasma triglyceride and LDL [13,14], a lower HDL cholesterol [14,15], and a lower HDL to total cholesterol ratio [16]. 

India is facing not only the largest number of type 2 diabetes patients in the world, but also a rapid rise in childhood obesity [17]. Studies consistently report that children of South Asian descent exhibit adiposity, insulin resistance, and metabolic disorders at a younger age compared to other ethnic groups [17,18,19]. South Asian adolescents are more insulin resistant, have more body fat, and have a higher risk of cardiovascular disease with higher blood pressure and fasting triglycerides [20] compared to white British adolescents. A strong gene–environment interaction has been suggested as the cause of the rapid increase in diabetes and metabolic syndrome in South Asians [4]. This condition is exacerbated by insufficient disease awareness and health care-seeking behavior, delay in diagnosis due to unusual symptoms and communication barriers, and socio-cultural as well as religious factors [21]. 

## 3. The Role of Body Composition 

The high prevalence of metabolic syndrome in South Asians may be partly explained by an unfavorable body composition, in which Asians have a higher body fat percentage than white Caucasians with the same BMI [22,23,24,25,26,27,28]. The difference in body composition was observed in both men and women. With the same BMI as Caucasians, South Asian men have a 4–7% higher body fat percentage [22,23,24,25,26], while women have an 8% higher body fat percentage [27,28]. Additionally, Asians have been consistently reported to have a lower fat-free mass (lean body mass) and/or appendicular skeletal muscle mass compared to other races [26,27,29], even after adjusting for height. Asians have the lowest lean mass index (fat-free mass index, FFMI) compared to other ethnic background such as Caucasians, African Americans, and Hispanics [30]. In South Asian men, the lean body mass was 3.4 kg lower than in Aboriginals, 3.0 kg lower than in Chinese, and 3.6 kg lower than in Caucasians [31]. In South Asian women, the lean body mass was 2.0 kg lower than in Aboriginals, 2.2 kg lower than in Chinese, and 3.0 kg lower than in Caucasians [31]. 

Interestingly, the unfavorable body composition is already present at a young age. South Asian adolescents of 14–17 years old [32], 11–12 years old [33], and children of 5–7 years old [34] had higher levels of body fat compared to their European counterparts in the UK. Body fat percentages were significantly higher in middle school South Asian children than in Caucasian, East Asian, African American, and Hispanic children living in the US [35]. A study by Stanfield et al. [36] demonstrated that South Asian infants in early infancy (6–12 weeks) had 0.34 kg less fat-free mass (FFM), and had an indication of a higher fat mass (FM) than white European infants. This difference persisted after adjustment for the smaller body size of South Asians. The study also found that for a given infant weight, the South Asian body composition shifted by 0.16 kg from FFM to FM. The differences in the amount of FFM were almost entirely explained by ethnic differences in the rate of growth in utero and length of gestation [36]. This finding confirmed the results of previous neonatal anthropometry studies comparing Indian babies born in India with white babies born in the UK, with a lower weight, smaller waist, and smaller mid-circumference, but more subscapular skinfold thicknesses have been observed as high in Indian babies [37,38]. Truncal adiposity was greater at the age of 4 y [38], suggesting a thin–fat phenotype from an early age. A longitudinal study conducted in a cohort in New Delhi, India [39], showed that birth weight and an increased BMI in infancy and early childhood were more predictive of lean mass in adults than adiposity in adults, whereas a greater increase in BMI in late childhood and adolescence predicted obesity in adults. This suggests that the postnatal environment may modify the development of an unfavorable body composition in South Asians. 

The estimation of body fat percentage based on BMI may not reflect the amount of atherogenic adipose tissue, i.e., visceral and ectopic fat adequately [40]. Fat distribution could be a more informative parameter. The sum of the truncal skinfolds in South Asian men was higher and was associated with a lower rate of glucose disposal [41] and a higher incidence of diabetes in women, but not in men [42]. Simple anthropometric indices such as waist circumference (WC), hip circumference (HC), waist to hip ratio (WHR), and skinfold thickness are useful measures to assess the risk of metabolic disease related to obesity in large population studies focusing on South Asians. However, these measurements are a proxy for an abdominal fat content (total abdominal fat: TAT; subcutaneous fat: SAT; visceral fat: VAT) and may differ in accuracy when ethnic groups are compared. Depending on the age and gender of the populations studied, some studies found differences in indices of abdominal adiposity (TAT, SAT, and VAT) when comparing South Asian and BMI-matched Caucasians, while others did not. Studies comparing a group of South Asian men and women of the same age, BMI, and WC with Caucasian men and women found greater abdominal adiposity in South Asians [22,43]. Despite similar WC, part of that study [22] also showed South Asians had an unfavorable lipid profile [44]. Visceral fat was found to mediate the effect of ethnicity on risk factors for developing dyslipidemia and CVD, suggesting that the high risk of CVD in South Asians may be attributed to higher visceral fat [45], whereas another study found a stronger correlation between CVD risk and subcutaneous fat [46]. A study of relatively older South Asian and Caucasian men with the same BMI found no difference in the WHR and visceral fat area, despite a higher body fat percentage in South Asian men [23]. Another study compared young South Asian and Caucasian men with the same BMI, and reported that the body fat percentage, subcutaneous fat, and adipocyte size were higher in South Asian men, but no difference in intra-peritoneal fat was found [24]. The subcutaneous abdominal compartment can be divided into superficial and deep subcutaneous; in this respect, South Asians had more deep subcutaneous fat than BMI-matched Europeans [47,48]. Interestingly, a study of young South Asian and white women showed no difference in any of the abdominal obesity measures and no metabolic disease was observed [49]. The discrepancy found in the studies may be attributed to the characteristics of the subjects, the matching procedure of the two ethnic groups, and the number of subjects. However, in large population studies [22,45,47,48], differences in the body fat distribution were observed between South Asians and white Caucasians towards a more central fat depot in South Asians. A gene–environment interaction of obesity-related traits was confirmed by a longitudinal genome-wide association study in a large cohort in Southern India, as indicated by an association between the rs9939609 variant at the FTO locus with adiposity measures (BMI, WC, HC, WHR, skinfold thickness) and metabolic consequences [50]. 

The unfavorable distribution of body fat towards a more centrally located fat depot may not be the only characteristic responsible for developing metabolic complications in South Asians. Adipose tissue serves as an energy store and as an endocrine organ that produces adipokines to regulate energy balance. It has been suggested that South Asians and Caucasians differ in adipokine production. Studies have found a decrease in adiponectin and an increase in leptin concentration in South Asians which contributed to a higher prevalence of diabetes [51,52]. Furthermore, a higher ApoB/ApoA-I ratio was found in centrally obese South Asians, suggesting that it may be a risk factor for developing CVD [53]. 

Finally, the susceptibility of South Asians to central adiposity and atherogenic dyslipidemia at a lower range of body fat than white Caucasians led to the hypothesis of adipose tissue expandability and lipid overflow [54]. South Asians have been suggested to have a smaller superficial subcutaneous adipose tissue compartment than Caucasians. As obesity develops, South Asians exceed the storage capacity of that compartment earlier than Caucasians, causing a lipid overflow into deep subcutaneous adipose tissue and into visceral adipose tissue, which, subsequently, leads to dyslipidemia [54]. The limited storage capacity of superficial subcutaneous adipose tissue may be characterized by a larger adipocyte size in South Asians than Caucasians, accounting for ethnic differences in insulin, HDL-cholesterol, adiponectin, and ectopic fat accumulation in the liver [55].

## 4. Energy Intake 

Developing countries are experiencing rapid dietary changes that are associated with a higher incidence of obesity and metabolic syndrome [56,57]. Diet-related factors are likely to increase the incidence of metabolic syndrome in South Asians already predisposed to obesity. The South Asian diet is characterized by a high carbohydrate intake (60–67%), SFA (from animal fat, coconut oil, palm oil, and *ghee* butter), TFA, n-6 PUFA (sunflower, safflower, corn, soybean, and sesame oil) and a low intake of n-3 PUFA (thus, a lower ratio of n-3/n-6 PUFA) and a low intake of dietary fiber [56,58]. An increased dietary n-6 PUFA and SFA intake in South Asians was found to be associated with fasting hyperinsulinemia and subclinical inflammation, respectively [59].

Immigration to developed countries has been shown to increase the risk of atherosclerosis in South Asians; the longer the time that passes since immigration, the greater the risk of developing atherosclerosis [60]. The body composition of South Asians shifted towards a higher body fat as the length of stay in the UK increased [61]. It has been suggested that dietary acculturation after migration to Western countries may be involved in the development of metabolic syndrome in South Asians [61]. The main nutritional change in South Asians after migrating to Norway was a substantial increase in energy and fat intake; a decrease in carbohydrate intake, especially a switch from complex to refined carbohydrates; increased consumption of meat and dairy products, and a reduced vegetable intake [62]. In the UK [63] and Canada [13], South Asians had a significantly higher carbohydrate intake (50% or more), associated with a lower HDL and a lower ratio of HDL to total cholesterol. In a 12-country study, the prevalence of low HDL was the highest in South Asians (63% in non-diabetic and 67% in diabetic) [15]. As the length of their stay in Canada increased, South Asians adopted a more positive dietary practice, that is, a higher consumption of fruits and vegetables and a lower consumption of fried foods; however, there has been a higher consumption of convenience foods, sugary drinks, and meat, and more frequent dining out [64], while another study found no difference in the eating patterns of multi-ethnic adolescents (including those of South Asians), suggesting acculturation [65]. 

Nutrient imbalances in the indigenous diet of South Asians natives, as well as dietary acculturation of South Asian immigrants in Western countries, can deteriorate long-term metabolic complications. In Caucasians, a high carbohydrate intake (mono- and polysaccharides) is associated with an elevated TAG, because of an increased TAG production (via increased de novo lipogenesis and VLDL synthesis) or decreased TAG clearance [66]. A reduction in HDL cholesterol has been observed in South Asians who consumed a large amount of carbohydrates in their diet [13]. Additionally, SFA and TFA intakes have been linked to an increase in LDL cholesterol [67]. Therefore, a combination of imbalanced nutrients in the South Asian diet may contribute to the excessive risk of CVD in this population. 

## 5. Physical Activity (Activity-Induced Energy Expenditure) 

Numerous epidemiological studies have been conducted to assess the physical activity level of South Asians predisposed to obesity. In fact, South Asian migrants in Western countries have been reported to have lower levels of physical activity than Caucasians [68,69,70,71]; likewise, this pattern has also been observed in adolescents [65]. In South Asians, less physical activity was associated with a greater adiposity, while greater physical activity levels were associated with a smaller waist circumference [68]. South Asians with higher visceral fat had less moderate physical activity, and moderate-to-vigorous physical activity compared to Caucasians; however, after adjusting to the levels of physical activity, visceral fat remained significantly higher [69]. In addition, vigorous activity was a predictor of liver fat accumulation, but moderate or moderate-to-vigorous activity may not be enough to prevent liver fat accumulation [72]. Given that South Asians are at an increased risk for cardio-metabolic disease, the question is whether the recommendation to perform moderate physical activity (MPA) of 150 min/week for Caucasians is sufficient for South Asians [73]. South Asians have been shown to need a higher amount of moderate physical activity (MPA) for 266 min/week to produce a cardio-metabolic risk profile similar to that of Caucasians [73]. 

Compared to Caucasians, South Asians also showed lower cardiorespiratory fitness and VO_2_max [71,74]. However, a lower cardiorespiratory fitness and VO_2_max in South Asians in these studies were not corrected for a difference in fat-free mass between ethnicities. Lower cardiorespiratory fitness, less physical activity, and a greater total adiposity together accounted for 83% of the ethnic difference in HOMA (IR), whereas 63% of the ethnic difference in fasting glucose was explained by cardiorespiratory fitness and total adiposity [71]. Cardiorespiratory fitness is closely related to the lipid oxidative capacity of skeletal muscle; South Asians have been shown to oxidize less fat during sub-maximal exercise, but there is no difference in fat oxidation during rest compared to Caucasians [74]. Interestingly, the lower oxidative capacity during submaximal exercise is not explained by a higher expression of oxidative and lipid metabolism genes in the skeletal muscle, but rather by a lower expression of insulin-signaling proteins in the skeletal muscle [74]. Mitochondria, as the main organelle involved in energy metabolism, had a higher oxidative phosphorylation (OXPHOS) capacity in diabetic and non-diabetic South Asian Indians compared to Caucasians with the same BMI [25]. Thus, mitochondrial dysfunction and oxidative gene expression in skeletal muscle may not explain the lower oxidative capacity during submaximal exercise in South Asians, that leads to the development of obesity and insulin resistance. Increased gene expression may not translate into increased protein (enzyme) synthesis, which is responsible for lipid metabolism. Furthermore, the oxidation of fatty acids can only tell one part of the story; the mobilization and handling of fatty acids and their storage are equally important. 

## 6. Total Energy Expenditure and Substrate Utilization 

Comparative studies have shown that Asians differ in their body composition from Caucasians when adjusting for BMI. Asians have a higher body fat percentage, whereas Caucasians have a higher fat-free mass (FFM) as reported by us [75] and others [23,24,25,26,27,28,76,77,78,79,80,81]. Among the three main ethnic groups in Asia, South Asian Indians had the most significant difference in body fat percentage compared to Caucasians, followed by Malay and Chinese [82]. Since there is a relationship between BMI and body fat for an ethnic group [83], differences in body fat percentage can be a confounding factor when comparing Asians and Caucasians with a similar BMI in an intervention study. By matching the two ethnic groups for body fat percentage rather than BMI [84], a misinterpretation of the response to diet or exercise can be avoided. 

The relationship between body composition and energy expenditure has been studied in health and diseases. Studies have established FFM as the main determinant of resting energy expenditure (REE) or resting metabolic rate (RMR) [85], which is the largest component (60–70% in moderately active adults) of total energy expenditure (TEE). Although RMR and also the sleeping metabolic rate (SMR) correlate best with FFM, RMR is also independently influenced by fat mass; the greater the FM, the higher the RMR [86]. In the study we conducted, Asians were found to have a lower RMR [29] and SMR than Caucasians because they have a lower FFM [75]. The difference in RMR and SMR disappeared with an adjustment for body composition [29,75]. 

A low RMR, after adjusting for a lean body mass, fat mass, age, and gender were associated with an increased risk of body weight gain in a 3-year follow-up study [87,88]. An unfavorable body composition, which is manifested in a greater accumulation of body fat, is possibly the consequence of a lower energy expenditure and a higher respiratory quotient (RQ), which indicates a lower proportion of fat-to-carbohydrate oxidation. Alternatively, low energy expenditure may be due to an unfavorable body composition. In a longitudinal study, a higher RQ was associated with a susceptibility to weight gain in Pima Indians [89]. However, Weyer et al. [90] reported there was no evidence for a lower metabolic rate or impairment in 24 h fat oxidation in Pima, who are predisposed to obesity, compared to Caucasians. Similarly, Bergouignan et al. [91] showed no differences in 24 h fat oxidation in obese, reduced obese, and lean subjects under a similar caloric and negative energy balance. Despite differences in body composition, a 24 h fat oxidation was similar between Asians and Caucasians when fed the same energy balance diet [75].

The 24 h fat oxidation reflects both the endogenous fat oxidation, mostly during the post-absorptive period, and the exogenous fat oxidation from dietary fats during the postprandial period. Most of the exogenous fat is stored during the postprandial period, while a limited fraction (~10%) is directly oxidized [92]. In the study by Wulan et al., fat oxidation was similar in Asians and Caucasians despite differences in body composition, being on average 11.7% and 10.4% of dietary fat, respectively [75]. Previous studies using a tracer showed that dietary fat oxidation in men and women was negatively correlated with body fat percentage [93], whereas others found a higher dietary fat oxidation with an increase in body fat percentage in men [94]. Interestingly, a study comparing obese, reduced obese and lean men and women and found no difference in the dietary fat oxidation [92]. The discrepancy in the results may be due to the differences in the metabolic fate of the dietary fatty acid tracers used, i.e., palmitic acid or oleic acid, the label, i.e., ^2^H or ^14^C, and the duration of the observation [92]. 

In the study by Astrup et al. [95] and Buemann et al. [96], substrate utilization was compared in lean subjects and post-obese subjects (predisposed to obesity) rather than obese subjects. In both studies, subjects were provided with isocaloric low- and high-fat diets. They found that formerly obese women failed to increase fat oxidation in response to increased dietary fat as compared to lean women. Substrate utilization was measured as the ratio of fat to carbohydrate oxidation [95] or postprandial fat oxidation after supper [96] and was independent of energy balance. 

Energy balance in the respiration chamber-predicted 24 h fat oxidation in Asians and Caucasians [75] and predicted 24 h RQ [89]. The more negative the energy balance, the higher the fat oxidation (the lower the RQ); the more positive the energy balance, the lower the fat oxidation (the higher the RQ). It is hypothesized that South Asians may have a lower capacity to oxidize fat from food, resulting in fat accumulation and an unfavorable body composition. Our study compared substrate utilization by South Asian and Caucasian men in response to overeating with a high-fat diet under sedentary conditions in a respiration chamber to avoid variability in physical activity levels. The study showed that the association between 24 h fat oxidation and energy balance was weak because all subjects had an extremely positive energy balance as a result of the combined effect of overeating and low physical activity [97]. 

Energy balance indicates whether there is a gap between energy intake and energy expenditure. Macronutrient oxidation in the body is determined by the need to regenerate ATP [30]; thus, equals TEE, which is the sum of the resting energy expenditure, diet-induced energy expenditure, and activity-induced energy expenditure. In this regard, nitrogen (protein) balance is easily maintained with a high or low (but sufficient) protein intake [98]. Likewise, the body’s ability to store glycogen is limited to a few hundred grams and, generally, remained within a relatively stable range [98]. Therefore, the oxidation of proteins and carbohydrates is adapted to the intake of proteins and carbohydrates, respectively; consequently, protein and carbohydrate balances are achieved. 

In contrast to the other two macronutrients, numerous studies on dietary fat supplementation or isoenergetic diets with a high- or low-fat content have shown that fat consumption does not promote its own oxidation; on the contrary, body fat stores are large [99]. The adaptation of fat oxidation to a higher fat intake in lean subjects consuming an isocaloric diet occurred in 7 days [100] and the adjustment of fat oxidation to match the fat intake could be accelerated when glycogen stores were reduced through strenuous exercise in both lean [101] and obese subjects [102]. This suggests that fat oxidation is regulated primarily by the use and storage of carbohydrates in the body or, in other cases, by the gap between total energy expenditure and energy ingested in the form of carbohydrates and proteins, rather than by the quantity of fat consumed in a particular day [99]. 

Isocaloric diets that differed in the composition of macronutrients (protein/carbohydrate/fat, in a mixed diet: 15/55/30; a high-fat diet: 15/25/60; a diet rich in carbohydrates: 15/70/15) showed a relative contribution of each macronutrient oxidation to TEE (substrate partitioning), close to the macronutrient composition [103]. Likewise, a reduction in the oxidation of dietary fat was observed after a diet rich in carbohydrates [104]. 

Our study showed that when South Asian and Caucasian men were overfed with a high-fat diet under sedentary conditions, an extremely positive energy balance was created and glycogen stores were relatively conserved (not depleted) because physical activity was low [97]. Therefore, although the diet was high in fat (60% energy intake), substrate use showed a greater dependence on carbohydrate oxidation in both South Asians and Caucasians, being on average 53% and 47% of TEE, respectively, whereas average fat oxidation in South Asians and Caucasians accounted for 31% and 39% of TEE, respectively [97]. In sedentary conditions (low physical activity), the need to replenish the body’s glycogen stores was low; therefore, most of the glucose oxidized to match the carbohydrate intake, whereas fat was oxidized to a lesser extent to meet energy requirements, resulting in a positive fat balance [97]. Thus, most of the fat in the diet was stored. A schematic representation of the mechanism of body fat accumulation in South Asians is shown in Figure 1, which includes various components of energy balance.

## 7. Molecular Adaptations in Adipose Tissue in Relation to Fat Metabolism in South Asian and Caucasian Men Overfed with a High-Fat Diet 

Our study on the molecular adaptation to overfeeding with a high-fat diet (under sedentary conditions) in South Asian and Caucasian men showed a significant decrease in short-chain 3-hydroxy acyl-CoA dehydrogenase (HADH) levels [104], a crucial enzyme for mitochondrial β-oxidation involved in the rate-limiting acyl CoA dehydrogenase step [105] in both ethnicities. The decrease in HADH can be expected in response to a massively positive energy balance, along with some metabolic alterations such as elevated insulin concentrations, and decreased fasting plasma NEFA [97], indicating the suppression of lipolysis, whereas the decrease in the concentration of fasting TAG indicates the uptake of fatty acids in adipose tissue. All together, they point towards the storage of fatty acids rather than oxidation. Contrasting results have been reported in caloric restriction studies, in which HADH levels increased [106,107]. Our study also showed that Caucasians had a relatively higher level of HADH at baseline and, while decreasing, the level of HADH after overfeeding remained higher than those in South Asians. This characteristic may be important in the long-term energy (fat) metabolism in this population. 

The entry of the acyl group into the mitochondrial matrix is catalyzed by the Carnitine palmitoyltransferase 1 alpha (CPT1a), which converts cytoplasmic long-chain fatty acyl-CoA to acylcarnitine [105,106]. Here, the study [104] showed that CPT1a changes significantly with diet, decreasing among South Asians but increasing among Caucasians. This may indicate a greater capacity for a mitochondrial fatty acids uptake in Caucasians. 

Perilipin (PLINA), which plays a role in the turnover of stored TAG and is found in lipid droplets [108,109] did not change significantly with diet, but the changes differed between ethnicities [104]. PLINA decreased more in Caucasians, whereas, before and after intervention with a high-fat diet, PLINA levels were slightly higher in South Asians, which may influence the long-term energy utilization in this population as well [104]. 

## 8. Conclusions

South Asians have an unfavorable body composition, which can be the cause or consequence of a low rate of energy metabolism. The causal relationship may be demonstrated in a longitudinal study. South Asians are predisposed to obesity, when exposed to an unhealthy lifestyle such as an increased energy intake or reduced physical activity leading to an earlier onset of obesity and metabolic syndrome. South Asians had a lower resting energy expenditure due to lower fat-free mass, and were less physically active. Thus, from an energy expenditure perspective, South Asians were more susceptible to a positive energy balance, in particular when they were exposed to a higher energy intake. In a condition of positive energy balance, energy metabolism favors glucose oxidation rather than fat oxidation. In this way, body fat is accumulated in the long-term through a chronically positive energy balance. 

Lifestyle interventions, such as the modification of macronutrients distribution in the diet to improve body composition in South Asians, are important. In practice, South Asians can benefit from an increased protein intake in exchange for a reduced carbohydrate and fat intake. Although physical activity has been prescribed to address the obesity problem, a different regimen of a longer duration or higher intensity physical activity may be recommended for South Asians. The higher energy expenditure from physical activity may partially compensate a lower resting energy expenditure, and reduce the risk of developing obesity. 

## Figures and Tables

**Figure 1 nutrients-13-03776-f001:**
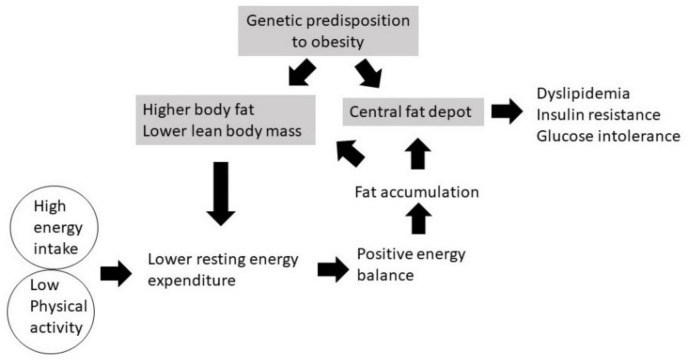
Mechanism of the development of obesity in South Asians.

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
