# Peer review of "Energy Metabolism in Relation to Diet and Physical Activity: A South Asian Perspective"

_nutrients, 2021, doi:10.3390/nu13113776_

Round 1

Reviewer 1 Report

The review written by Siti N. Wulan and colleagues describes the issues of obesity, overweight, metabolic syndrome from the perspective of the Asian population.

The latter shows differences compared to the Western countries in terms of body composition, fat distribution, energy expenditure, fat metabolism and others. The review describes an interesting topic and an interesting perspective; however, a major revision should be considered.

First, the English should be revised.

Secondly, I would appreciate to see a reorganization of the sections. It should be clear from the introduction how the review is structured, and which points the authors intend to cover. The authors are focusing on different topics, such as the role of the body composition, the imbalance between energy intake and energy expenditure, and the physical activity. A possible reorganization of the text could be achieved by considering the following sections:

  • Body composition (including actual section 3)
  • Physical activity (including actual section 5)
  • Energy intake (including actual section 4)
  • Energy expenditure/Energy usage by the body (including actual section 6, 7, 8)

The conclusion section and the take home message are not clear. It should be reformulated.

Finally, a section on the possible strategies that the Asian population can undertake to ameliorate the situation would be appreciated.

Author Response

  1. First, the English should be revised.

Answer: We tried to revise the English and style of the sentences, as shown in the track changes document.

  1. Secondly, I would appreciate to see a reorganization of the sections. It should be clear from the introduction how the review is structured, and which points the authors intend to cover. The authors are focusing on different topics, such as the role of the body composition, the imbalance between energy intake and energy expenditure, and the physical activity. A possible reorganization of the text could be achieved by considering the following sections:

Body composition (including actual section 3)

Physical activity (including actual section 5)

Energy intake (including actual section 4)

Energy expenditure/Energy usage by the body (including actual section 6, 7, 8)

Answer:

  • How the review is structured and the points we want intend to cover have been added in the introduction section (shown in track changes document, or highlighted in grey in clean version document)
  • The possible reorganization has been addressed, with the physical activity section is placed after the energy intake section. This is because physical activity is part of energy expenditure

  1. The conclusion section and the take home message are not clear. It should be reformulated.

Answer: The conclusion section and the take home message has been reformulated

  1. Finally, a section on the possible strategies that the Asian population can undertake to ameliorate the situation would be appreciated.

Answer: The possible strategies for Asian population has been added in the paragraph 2 of the conclusion section

Reviewer 2 Report

The purpose of this review was to highlight the impact of diet and physical activity on the surging obesity statistics among Asian populations.  The authors present that Asians have higher body fat compared to BMI matched Caucasians, differences in fat distribution and storage compared to Western populations, lower physical activity levels, and lower resting energy expenditure. Collectively, this leads to an increased risk of cardio-metabolic disease.  The authors suggest that differential expression of key enzymes in fat metabolism and mitochondrial function between Asians and other ethnicities, at least partly explain the rapidly increasing numbers of obesity and diabetic Asians.

Strengths

1)Very well written

2)Excellent detail regarding comparisons of important physical activity and cardio-metabolic data

3) Very balanced

Improvements

1) A figure illustrating the prevalence data would be helpful

2) A figure illustrating differences in fat distribution and impact on health outcomes would be helpful

3) The conclusion section would benefit from the author's proposing lifestyle modifications to improve physical activity and dietary habits for the Asian population that would help reduce the obesity rate.  

Author Response

  1. Strengths

1)Very well written

2)Excellent detail regarding comparisons of important physical activity and cardio-metabolic data

3) Very balanced

Answer: The authors would like to thank for the appreciation

  1. Improvements

1) A figure illustrating the prevalence data would be helpful

 Answer: A figure illustrating the prevalence of overweight and obesity has been added to the manuscript under the introduction section (Figure 1)

2) A figure illustrating differences in fat distribution and impact on health outcomes would be helpful

Answer:   The figure illustrating differences in fat distribution and impact on health outcomes has   been provided (Figure 2)

3) The conclusion section would benefit from the author's proposing lifestyle modifications to improve physical activity and dietary habits for the Asian population that would help reduce the obesity rate.  

Answer: Our proposing lifestyle modifications to reduce obesity rate in Asians have been added

               in the conclusion section

Round 2

Reviewer 1 Report

I appreciated the changes and improvements the authors made on the manuscript.

However, there are still some points which I would like to highlight:

  • The first newly written paragraph (reported here below in red) is way too long. It should state concisely the aim of the review without adding further information which will be included in the later section. Please shorten it. “ This review addressed the possible determinants of the increased prevalence of obesity and metabolic syndrome in South Asian population. The obesity in South Asians was characterized by an unfavorable body composition i.e. a higher body fat percentage and a lower fat-free mass/lean body mass and an unfavorable body fat distribution towards a more centrally fat depot for a given BMI as whites. It is hypothesized that the unfavorable body composition i.e. a higher body fat is a result of a lower capacity of fat oxidation in South Asians, alternatively the unfavorable body composition i.e. a lower fat-free mass lead to a lower (resting) energy expenditure and a lower capacity of fat oxidation, mediated by the susceptibility towards a more positive energy balance. Studies in young age South Asians suggested that this population predispose to obesity. The inability of this population to adapt to the in- creased energy (fat) intake dan reduced physical activity may contribute to the rapid in- crease of obesity and metabolic syndrome in South Asian population. “
  • Figure 1. It comes out of the blue. It is not mentioned in the text and it misses the description of the axes. Is it needed after all?
  • The Conclusion section is better presented; however, it still should not include “results”, it should rather summarize what it is described in the previous sections. Please revise it.
  • Possible strategies that the Asian population can undertake to ameliorate the situation were included in the “Conclusion” section. The authors could also consider writing a separate section before the Conclusion.
